# ID2 Promotes Lineage Transition of Prostate Cancer through FGFR and JAK-STAT Signaling

**DOI:** 10.3390/cancers16020392

**Published:** 2024-01-17

**Authors:** Jinxiong Zhang, Zhihao Chen, Yongxin Mao, Yijun He, Xin Wu, Jianhong Wu, Lu Sheng

**Affiliations:** Department of Urology, Huadong Hospital Affiliated to Fudan University, Shanghai 200040, China; 21211280011@m.fudan.edu.cn (J.Z.); 21211280034@fudan.edu.cn (X.W.)

**Keywords:** ID2, prostate cancer, lineage transition, ID2 UP50 signature, FGFR signaling

## Abstract

**Simple Summary:**

Androgen receptor-negative prostate cancer is one of the important mechanisms of castration resistant prostate cancer. The application of next generation androgen receptor signaling pathway inhibitors (ARPIs) has led to a gradual increase of AR-negative prostate cancer in the clinic. In this paper, we demonstrated the potential promotional role of ID2 in androgen receptor-negative prostate cancer through extensive bioinformatics analysis and experimental studies. Through transcriptome sequencing and downstream functional enrichment analysis, we found that ID2 can activate neuroendocrine or stemness-related pathways and inhibit androgen receptor signaling pathways. ID2 can activate JAK/STAT signaling pathway as well as FGFR signaling pathway to promote the acquisition of prostate cancer lineage plasticity, which in turn leads to androgen receptor-negative prostate cancer. Through cell function experiments and mouse experiments, we reveal that ID2 can promote prostate cancer evolution. Using transcriptome sequencing and publicly available clinical data, we generated ID2-related gene signatures to help determine clinical prognosis.

**Abstract:**

The use of androgen receptor pathway inhibitors (ARPIs) has led to an increase in the proportion of AR-null prostate cancer, including neuroendocrine prostate cancer (NEPC) and double-negative prostate cancer (DNPC), but the mechanism underlying this lineage transition has not been elucidated. We found that ID2 expression was increased in AR-null prostate cancer. In vitro and in vivo studies confirmed that ID2 promotes PCa malignancy and can confer resistance to enzalutamide in PCa cells. We generated an ID2 UP50 signature, which is capable of determining resistance to enzalutamide and is valuable for predicting patient prognosis. Functional experiments showed that ID2 could activate stemness-associated JAK/STAT and FGFR signaling while inhibiting the AR signaling pathway. Our study indicates a potentially strong association between ID2 and the acquisition of a stem-like phenotype in adenocarcinoma cells, leading to resistance to androgen deprivation therapy (ADT) and next-generation ARPIs in prostate cancer.

## 1. Introduction

Prostate cancer (PCa) is the second most prevalent cancer among men and the fifth leading cause of cancer-related deaths [1]. Androgen deprivation therapy (ADT) has been established as the standard treatment for metastatic PCa [2]. However, the efficacy of ADT varies among patients, and most patients eventually develop castration-resistant prostate cancer [3]. The use of next-generation AR pathway inhibitors (ARPIs) prolongs overall survival and PSA progression-free intervals in CRPC, but drug resistance still occurs after a period of ARPI administration [4,5,6,7]. Following the widespread clinical use of next-generation ARPIs, the proportion of AR-null prostate cancer, including neuroendocrine prostate cancer (NEPC) and double-negative prostate cancer (DNPC), has subsequently risen as a means of drug resistance [8].

NEPC histologically resembles small cell lung cancer (SCLC): small cells with deeply stained nuclei and increased nucleoplasmic ratio. Common immunohistochemical diagnostic markers for NEPC include CHGA, SYP, NSE, and NCAM1, which are positive, while AR is negative [9]. However, not all NEPCs demonstrate positive NE markers. Unlike de novo NEPC, treatment-induced NEPC (t-NEPC) is derived from adenocarcinoma of the prostate (ADPC), as numerous studies have shown that t-NEPC shares almost the same genomic alterations as ADPC (TMPRSS2-ERG fusion) [10,11,12,13]. Mechanically, the current research indicates that the transdifferentiation of adenocarcinoma can be categorized into two stages: firstly, the tumor acquires lineage plasticity under the stress of ARPIs; secondly, the tumor transdifferentiates into an NEPC, accompanied by the loss of reliance on the androgen signaling pathway [14]. This process involves numerous molecular events. At the genomic level, TP53 and RB deletions exhibit significant prevalence in NEPC in comparison to ADPC. RB1 loss is typically noticeable in about 10% of cases in metastatic CRPC and is linked to poorer prognoses [11]. In addition, the concurrence of PTEN deletion, TP53 mutations, and RB1 loss is associated with lineage plasticity and neuroendocrine prostate cancer (NEPC), characterized by its highly resistant response to treatment [11,15]. ASCL1 and MYCN amplification are also involved in neuroendocrine transdifferentiation (NETD) [13,16,17,18]. Epigenetic modifications in NEPC, including histone methylation and acetylation, as well as DNA methylation, are pervasive [14]. In NEPC, the histone methyltransferase EZH2 is highly upregulated, which results in an increased distribution of H3K27me3 and promotes the lineage plasticity of PCa [17]. During the acquisition of lineage plasticity in PCa, lineage-related pathways, such as epithelial–mesenchymal transition (EMT), WNT/β-catenin, JAK/STAT signaling, and calcium signaling, are activated [19,20,21,22]. Currently reported NEPC drivers comprise ONECUT2, ASCL1, MYCN, PEG10, BRN2, SOX2, and HP1α [16,18,23,24,25,26,27]. While the molecular characteristics of ADPC and NEPC are relatively well understood, those of DNPC are not established [28]. The current research shows that FGFR signaling may help AR-null prostate cancer bypass AR signaling [28].

ID2 has been widely researched in various diseases, and its molecular function is linked to cellular growth, senescence, differentiation, apoptosis, angiogenesis, and neoplastic transformation. In glioblastoma, the oncoprotein N-Myc increases ID2 expression and binds with Rb, resulting in its deactivation and subsequently promoting cell cycle progression [29]. Dephosphorylated ID2 decreases self-degradation, resulting in increased intracellular levels and promoting the proliferation of neural precursor cells [30]. ID2 decreases intracellular reactive oxygen species (ROS) production through the inhibition of oxidative phosphorylation in the mitochondria, thus promoting tumor cell survival in glioblastoma [31]. ASCL1 upregulates the expression of ID2 in SCLC [32]. However, ID2’s role in NEPC has not been reported in any study. Given the comprehension of the molecular mechanisms involved in NETD and the significance of ID2 in various cellular biological processes, we carried out a study to examine the function of ID2 in the process of the lineage transition of prostate adenocarcinoma.

## 2. Results

### 2.1. ID2 Expression Is Upregulated in NEPC and DNPC

To explore ID2 expression levels across different prostate cancer cell lines, we mined RNA-seq data from CCLE. The transcriptional sequencing data of these four cell lines were obtained from the CCLE database to compare the expression levels (TPM) of ID2. We found that ID2 transcript levels were lowest in the LNCAP cell line, which is a hormone-sensitive cell line, whereas in PC3 and 22RV1 (hormone-resistant cell lines), ID2 expression levels were elevated, and the highest levels of ID2 expression were found in NCI-H660, a de novo small cell neuroendocrine prostate cancer cell line [33] (Appendix A). We next performed a dual immunofluorescence analysis of ID2 and CHGA on samples from clinical neuroendocrine carcinoma and adenocarcinoma of the prostate, with the former being positive for CHGA and strongly positive for ID2, whereas in the latter, it was negative for CHGA and weakly positive for ID2 (Figure 1A and Appendix A). We then mined the expression of ID2 in Beltran’s data [13], and the expression level of ID2 in CRPC-NE was significantly higher than that in CRPC-AD (Figure 1B).

Given that only a fraction of PCa after ARPI administration acquired resistance through NETD, we investigated the differences in the altered expression levels of ID2 in NETD PCa and other PCa that acquired resistance through other mechanisms. We mined RNA sequencing data from 21 paired samples after enzalutamide treatment, of which 3 samples had histologically confirmed transdifferentiation to neuroendocrine prostate cancer (NEPC), while the remaining 18 did not develop NETD [34]. In the three samples with NETD, ID2 expression levels were elevated (Figure 1C), while in the remaining eighteen samples without NETD, ID2 expression levels were either elevated or decreased, with nonsignificant differences.

To determine whether a difference in the state of chromatin in the ID2 genomic region in NEPC and ADPC exists, we mined CHIP-seq data for H3K27ac from a series of LuCaP PDX mice and found that NEPC has a more significant H3K37ac signal peak near the ID2 gene (Figure 1D). H3K27ac is a histone modification that enhances gene expression, and this epigenomic characteristic in NEPC is consistent with the above results of elevated transcript levels of ID2 in NEPC [35].

To explore the differences in pathways in ID2-high-expressing tumors, 499 TCGA tumors were subjected to GSEA and grouped based on the median expression level of ID2, which showed that the activity of neurological pathways such as neuron projection guidance and the regulation of neurogenesis was higher in the high-expressing group (Figure 1E).

To determine the expression of ID2 in DNPC, which is AR-negative and NE-negative, we mined RNA-seq data from Labrecque and SU2C2019. The results show that transcripts of ID2 are significantly upregulated in DNPC (Figure 1F).

### 2.2. Transcriptomic Reprogramming and Pathway Analysis upon ID2 Overexpression

We first investigated whether ID2 induces changes in gene expression patterns. According to PCA analysis, PC3 cells with ID2 overexpression showed differences from the control group in terms of principal components. The findings revealed that ID2 leads to alterations in cellular transcription patterns (Figure 2A). Differential analysis was performed using edgeR. In total, 865 differential genes were identified, of which 772 were upregulated genes, and 93 were downregulated genes (Figure 2B). The result of the differential analysis is shown in Appendix A. We analyzed the expression of the top 20 differentially expressed genes (DEGs) in a panel of PCa cell lines (Figure 2C). The heatmap demonstrates that these DEGs are moderately higher-expressed in NE, like PC3, DU145, and NCI-H660 cells. The most differentially upregulated genes were IFI6 and IFI27, which play an important role in apoptosis and may promote the growth and migration of tongue squamous cell carcinoma [36]. Notably, we also found a significant downregulation of HOXB13, an androgen co-stimulatory factor, the absence of which leads to lipid accumulation in prostate cancer cells, which increases cell motility and causes metastasis in xenograft tumors [37]. In addition, we stably transfected ID2 in LNCAP cells and cultured it with charcoal-stripped serum; we found that cells underwent axon generation, and HOXB13 expression was downregulated (Figure 2D,E).

Next, we aimed to know whether ID2 activates or inhibits certain pathways in PCa cells, which, in turn, influences its biological function. We performed GSEA using hallmark pathways from MsigDB and found that a total of 20 pathways were significantly activated or inhibited (Figure 2F). Immune-related pathways, such as IFN-α and IFN-β inflammatory responses, were the most significantly activated in the ID2-overexpression group and stemness-related pathways, including IL-6/Jak/Stat3 signaling and TGF-β signaling. This finding was consistent in the activated hallmark pathways with genetically engineered mouse models (GEMMs) that recapitulate the lineage transition [38].

Most transcription factors exert their function through their modifier status without changing their expression as RNA-seq data fail to capture post-transcriptional and post-translational modifications. Therefore, we conducted a master regulator analysis to identify changes in the major TF activity after ID2 overexpression, which identified 69 activated master regulators (Appendix A). The top activated transcription factors are HMX1, TLX1, and CRX (Figure 2G). To confirm whether these activated master regulators play a more critical role in clinical de novo NEPC than ADPC, we conducted expression analysis and identified 1225 DEGs (Appendix A). Then, we took the intersection of these differentially expressed genes and the above master regulators and identified four genes, of which ASCL1 and NKX2-1 have been confirmed to be NETD drivers by related studies [35] (Appendix A). Enrichment analysis of these activated regulators shows activation in the pathways associated with nervous system development and endocrine development (Figure 2H).

### 2.3. ID2 Attenuates AR Signaling and Promotes NETD

Based on the fact that the AR pathway is less active in NEPC [39], we then aimed to determine whether ID2 will exert a role on that. We transiently expressed ID2 in a panel of prostate cancer cell lines; Western blot analysis showed decreased expression of AR and PSA in LNCAP cells, while the expression of NE markers such as CHGA, ENO2, and SYP was elevated in both LNCAP and PC3 cells (Figure 3A). Immunofluorescence (IF) analysis showed a significant decrease in AR after overexpression in C42B cells (Figure 3B). In 22RV1 cells, we knocked down ID2, and the qPCR showed that PSA and AR expression level was decreased, indicating decreased activity of AR signaling (Figure 3C).

Next, we performed transcriptome sequencing analysis of ID2-overexpressing PC3 cells, and GSEA shows that the androgen response pathway was suppressed (Figure 3D). We noticed that HOXB13 expression is downregulated in the transcriptomic data, which is an AR co-factor, and highly expressed in benign prostatic tissues and ADPC, while expression is decreased or absent in NEPC (Appendix A) and is reported to accelerate NETD [40]. On the premise that calcium signaling is activated in NEPC [22], we then examined the calcium level in C42B cells, and IF analysis shows a significant increase after ID2 overexpression (Figure 3E).

In addition, we evaluated the NE scores of 499 primary prostate adenocarcinoma cases in the TCGA database. Using Beltran’s NE gene set [13], a single-sample gene set enrichment analysis (ssGSEA) was performed on prostate tumor samples, with higher scores indicating a higher likelihood of NETD, and then, the NE score of each sample was correlated with the ID2 expression level. We found that the NE score of the samples had a significant positive correlation with the expression of ID2 (Figure 3F). Similarly, the SU2C2019 data also showed a significant positive correlation between the ID2 and NE scores and a significant negative correlation with AR pathway activity, as assessed by the five-gene score [41] (Figure 3F). In addition, we obtained the AR-repressed signature and RB-loss up signature and then used the signature to perform ssGSEA in our RNA-seq data to determine the signature scores of the samples. The results show that in the PC3 ID2-OE group, the AR-repressed signature and RB-loss up signature scored higher (Figure 3G).

To determine whether ID2 promotes tumor growth in vivo, we transplanted long-term ID2-overexpressing PC3 cells and control cells into nude mice; the tumor volume growth curve showed that ID2 can increase the growth rate of PCa (Figure 3H).

### 2.4. ID2 Enhances PCa Cell Aggressiveness

To investigate the effect of ID2 on the common hallmark phenotype of PCa cells, we first transiently expressed ID2 in PC3 cells. CCK8 experiments showed that ID2 could promote the proliferation of PC3, while in LNCaP cells, ID2 inhibited its proliferation (Figure 4A,B). This seemingly contradictory result was consistent with the above results that ID2 could inhibit the AR signaling pathway. As LNCaP is a hormone-sensitive cell line, ADT can inhibit its proliferation, whereas PC3 is a hormone-resistant cell whose growth does not depend on the androgen signaling pathway. However, after the long-term overexpression of ID2 with ADT, this inhibitory effect could be reverted. In the meantime, it can also eliminate cellular contact inhibition and confer resistance of enzalutamide to LNCaP cells (Figure 4C). Similarly, the transient expression of ID2 also had different effects on the migratory ability of different PCa cell lines. It inhibited the migratory ability of LNCaP but increased the migratory ability of PC3 cells (Figure 4D,E). However, ID2 could enhance the invasive ability of LNCAP cells, which was consistent with the results of the invasion assay of PC3 (Figure 4F). Meanwhile, we also demonstrated the role of ID2 in terms of EMT. Western blot showed reduced E-cadherin expression and increased β-catenin expression in ID2 stably overexpressing LNCaP cells cultured in charcoal-stripped serum (CSS) median (Figure 4G), indicating they can promote epithelial–mesenchymal transition. Consistently, the ssGSEA scores of the AR-repressed signature and RB-loss up signature were elevated in ID2-overexpressing PC3 cells.

### 2.5. ID2 UP50 Signature Generation and Validation of Its Effectiveness

Next, we focused on the genes most significantly upregulated by ID2. We generated an ID2 UP50 signature by incorporating the top 50 upregulated DEGs (Appendix A). This signature trended upward in CRPC and de novo NEPC cell lines versus HSPC (Appendix A), which further suggests that ID2 can confer drug resistance through NETD. To test its competence to infer NETD probability, we performed ssGSEA on samples from SU2C2019 and TCGA.

We determined whether this signature could classify NEPC and ADPC by calculating the signature score in samples with and without NE features. The SUC2C cohort does not show a significant difference in the ID2 signature score in NE and the adeno phenotype (Appendix A); we hypothesize that some of the adeno samples may have been in the early stage of NETD because of drug administration; thus, its transcriptomic profile has changed (from canonical adenocarcinoma morphology to small cell NEPC) but its terminal histological transformation has not. Instead, we performed correlation analysis with the ID2 signature score and NEPC score in the SU2C cohort and found the ID2 signature score is negatively correlated with the AR score and positively correlated with the Beltran NEPC UP score (Figure 5A,B).

Previous studies have shown that loss of RB1 and TP53 is more frequent in NEPC compared to ADPC, but this genomic event is not an obligatory drive to neuroendocrine transdifferentiation, as a subproportion of RB1−/− TP53−/− CRPC loci does not show an increase in NE markers, such as SYP and CHGA. We thus performed expression analysis in LNCAP DKO models. The results show that the ID2 UP50 signature score is not significantly changed in RB or TP53 single-knockout LNCAP cells, but after RB and TP53 double knockout, it was significantly raised concomitant with EZH2 expression, a known NEPC driver (Figure 5C and Appendix A). In addition, we also confirmed that ID2 can promote cell cycle progression (Figure 5D).

PCa can acquire enzalutamide resistance through different mechanisms, as it appears to show heterogeneous effects on transcriptomic profiles, which can be characterized by different clusters that represent different phenotypes [34]. We acquired the RNA-seq data of matched samples that underwent NETD and found the signature score is significantly upregulated after NETD (Figure 5E). In another cohort that underwent neoadjuvant therapy of enzalutamide, the signature score was also elevated after treatment (Figure 5F). In addition, this signature could also shed light on whether patients will respond to enzalutamide, as it scored higher in enzalutamide nonresponders than responders (Figure 5G)

To explore whether the ID2 UP50 signature can predict the clinical outcomes of PCa patients, we acquired clinical data from Abida [42] and Alumkal [43] and then calculated each patient’s sample signature score and defined patients with top decile or median scores as the exposure group. Survival analysis showed that the exposure group has an unfavorable clinical outcome (Figure 5H,I).

### 2.6. ID2 Activates FGFR Signaling and JAK-STAT Signaling to Promote Lineage Transition

Bluemn’s research demonstrates that AR-null prostate cancer is sustained through FGFR signaling [28]. Here, we explored the status of FGFR signaling in a panel of 20 prostate cancer cell lines. By clustering them into three phenotypes according to AR activity and NE activity, we found that FGF signaling is significantly activated in AR-null prostate cancer cells, including DNPC and NEPC (Figure 6A). Next, we examined FGFR signaling changes after ID2 overexpression. Our data showed that FGFR signaling is more active upon ID2 (Figure 6B). Given that JAK-STAT activation confers stemness to prostate adenocarcinoma [21,44], we performed GSEA on our data using the associated gene set and found it was activated after ID2 overexpression (Figure 6C). To further verify FGFR signaling and JAK-STAT signaling activation, we performed ectopic expression in LNCaP cell lines. The results of qPCR and Western blot show consistency in LNCaP and PC3 cells (Figure 6D,E). We next examined FGFR and JAK-STAT activity in clinical samples from TCGA and found that it was positively correlated with the ID2 level (Figure 6F). The addition of zoligratinib, an FGFR pathway inhibitor, attenuates the promotion of cell migration by ID2 overexpression (Figure 6G).

## 3. Discussion

The application of a new generation of ARPIs has improved the prognosis of patients on the one hand, but on the other hand, it has led to an increase in the clinical rate of AR-null prostate cancer, a refractory pathological type of PCa that is ineffective against commonly used androgen deprivation therapies. The current main treatment for AR-null prostate cancer is platinum-based chemotherapy, which is rapidly becoming resistant.

The gene level, epigenetic and transcriptional, and protein levels are all involved in the transdifferentiation process of PCa and are highly heterogeneous. Therefore, the discovery of universal mechanisms is crucial. Through multiple RNA sequencing datasets, we found elevated transcript levels of ID2 in AR-null PCa and confirmed it at the protein level through our clinical specimens. In addition, we also mined the epigenomic CHIP-seq of H3K27ac data, and the results show that the ID2 genome was also more active in NEPC. Next, we explored the altered intracellular transcriptional patterns induced by ID2 using RNA sequencing to explore the mechanism of lineage transition, and enrichment analyses showed that the pathway activated by ID2 was associated with neurodevelopment and that the AR pathway was repressed. Master regulator analysis suggested the elevated protein activity of ASCL1 and NKX2-1, which are transcription factors for neural spectrum development. Activation of the JAK/STAT pathway suggests that ID2 can also promote lineage plasticity in PCa. There are numerous studies showing the presence of a higher proportion of RB1 and P53 mutations in NEPC. It has been demonstrated that in glioblastoma, ID2 can bind to RB1 and thus inhibit its function [29]. The results of our cycle experiments in LNCAP cells show that ID2 can inhibit the cell cycle, so how ID2 inhibits the cycle and whether it affects the cycle by binding to AR or to RB1 needs to be further confirmed by subsequent studies.

Phenotypic studies show that ID2 has different effects in several PCa cell lines, and we speculated that such differences might be due to variations in the biological background of the cells. LNCaP is an androgen-dependent cell, and our pathway analyses show that ID2 can inhibit the AR pathway. The results of the CCK8 experiments show that the growth of LNCaP was inhibited by ID2, and the ability of the cells to migrate was relatively decreased. However, the invasive ability of LNCaP was slightly increased. As for the PC3 cell, an androgen-independent cell that is insensitive to enzalutamide, ID2 increases its malignancy in terms of cell proliferation, migration, and invasion. In vivo experiments also showed that ID2 can promote tumor growth.

Finally, we generated an ID2 signature based on DEGs and verified its effectiveness in several ways. We confirmed that the ID2 signature scored higher in the NEPC sample than in the ADPC sample. Correlation analysis showed that the ID2 signature was positively correlated with the NEPC signature and negatively correlated with the AR signature. Through clinical patient survival data and the corresponding sequencing sample data, we verified that patients with a high ID2 signature score have an unfavorable clinical outcome.

## 4. Methods

### 4.1. Cell Culture

LNCaP, 22RV1, C42B, and PC3 cell lines were obtained from the American Type Culture Collection (ATCC). LNCaP clones were cultured in RPMI-1640 medium supplemented with 1% penicillin and 10% fetal bovine serum (FBS) or 5% charcoal-stripped serum for ADT. 22RV1, C42B, and PC3 cell lines were cultured in RPMI-1640 medium supplemented with 1% penicillin and 10% fetal bovine serum (FBS). 293T cells were cultured in DMEM. For in vitro ARPIs, cells were treated with 10 μmol/ENZ and harvested at the indicated time points. Drugs were all obtained from MCE (China).

### 4.2. Animal Studies

All animal studies received prior approval from the IACUC of Washington State University and complied with IACUC recommendations. Male 4- to 6-week-old nude mice were housed in the animal research facility. First, 1 × 10^6^ ID2-overexpressing PC3 cells or control cells were injected subcutaneously into the flank of nude mice. Each group contained 8 mice. Tumor size was measured every 2–3 days using caliper after the tumor was formed. Tumor volume was calculated as length × width 2/2.

### 4.3. Western Blot

Total proteins were extracted from cells using RIPA buffer (Cat# P0013C, Beyotime, Haimen, China) supplemented with protease inhibitor (1:100, Cat# ZJ101, EpiZyme Biotechnology, Shanghai, China). Protein concentration was measured using BCA Protein Assay Kit (Cat# 23225, Thermo Fisher Scientific, Waltham, MA USA). Equal amounts of protein were separated using electrophoresis on 10% or 15% polyacrylamide gels (Cat# PG112, PG114 EpiZyme Biotechnology, Shanghai, China) according to protein mass, transblotted onto 0.22 nm or 0.45 μm polyvinylidene difluoride membranes, and incubated with antibodies against ID2 (1:1000, Cat# 3431S, Cell Signaling Technology, Danvers, MA, USA), FLAG (1:1000, Cat# AF519, Beyotime), GAPDH (1:1000, Cat# sc-365062, SANTACRUZ, Santa Cruz, CA, USA), SYP (1:1000, Cat# sc-365488, SANTACRUZ), ENO2 (1:1000, Cat# 24330T, Cell Signaling Technology), CHGA (1:1000, Cat# 60893S, Cell Signaling Technology), EZH2 (1:1000, Cat# 5246S, Cell Signaling Technology), AR (1:1000, Cat# ab108341, Abcam), PSA (1:1000, Cat# 5365T, Cell Signaling Technology), E-cadherin (1:1000, Cat# sc-8426, SANTACRUZ), N-cadherin (1:1000, Cat# sc-8424, SANTACRUZ), Vimentin (1:1000, Cat# D21H3, Cell Signaling Technology), Snail (1:1000, Cat# C15D3, Cell Signaling Technology), HOXB13 (1:1000, Cat# TA364479S, Origene, Rockville, MD, USA), and RB1 (1:1000, Cat# 9313T, Cell Signaling Technology) at 4 °C overnight. Afterward, the membranes were incubated with HRP-conjugated anti-rabbit IgG (1:3000, Cat# 7074S, Cell Signaling Technology) or anti-mouse IgG (1:3000, Cat#7076S, Cell Signaling Technology) secondary antibodies for 1 h at room temperature. In some experiments, the previous primary antibody and secondary antibody were stripped from the PVDF membrane using stripping buffer (Cat# PS107S, EpiZyme Biotechnology, Shanghai, China) for 10~15 min. After being re-blocked, the membrane was re-incubated with another primary antibody at 4 °C overnight. The next steps were the same as described above.

### 4.4. Cell Proliferation Assay

CCK8 (Cat# BR3001507) assay was applied to measure cell viability. In total, 2000 or 4000 cells were seeded into 48- or 96-well plate and cultured as indicated in the figures. At 0, 24, 48, 72, and 96 h, 10 μL or 20 μL CCK-8 working solution was added to each well and incubated for 2 h at 37 °C. Optical densities were then measured at a wavelength of 450 nm.

### 4.5. Cell Invasion Assay

Cell invasion was assessed by utilizing Matrigel-coated BioCoat Cell Culture Inserts (24-well plates, Corning, Corning, NY, USA). Once Matrigel had been rehydrated at room temperature, 2 × 10^5^ cells suspended in 200 μL of RPMI-1640 medium were seeded into each insert. At the base of each well, 500 μL of medium containing 20% FBS was added. After 48 h of culture, noninvading cells were removed, and the cells on the lower side of the membrane were subsequently stained with crystal violet.

### 4.6. Cell Migration Assay

Cell migration was assessed in a wound-healing assay. Cells were plated on 6-well plates and incubated for 24, 48, or 72 h after wound scratching. Wound confluence was captured at different time points using NIS-Elements Viewer. Wound closure was measured using ImageJ by comparing the mean relative wound density of three replicates.

### 4.7. RNA-Seq and Data Analysis

Total RNA of PC3 and LNCaP cells was extracted using RNeasy Mini Kit (74106, Qiagen, Venlo, The Netherlands) following manufacturer’s procedure. Whole transcriptome sequencing libraries were prepared using TruSeq^®^ Stranded mRNA Library PrepKit (RS-122-2101, Illumina, San Diego, CA, USA) by following the manufacturer’s instructions. RNA-Seq libraries were sequenced on a HiSeq2500 at Princess Margaret Genomic Centre. The trimmed reads were aligned to human genome hg38 with HISAT2, and gene expression was then quantified using the fragments per kilobase per million mapped reads (FPKM) method using Cufflinks (version 2.1.1) with GENCODE v24 GRCh37 GTF file. RNA-seq data is provided in Appendix A.

Differential analysis was performed using edgeR package, and DEGs were defined as |log Foldchange| > 1 and adjusted *p*-value < 0.05. GSEA was performed using the hallmark gene sets from version 4.0 of the molecular signature database (MSigDB). Master regulator analyses were performed by comparing ID2-overexpressing samples and control samples with the MARINa algorithm implemented in viper R package. The regulon was acquired through aracne.networks R package. (version: 1.24.0)

### 4.8. Signature Score Calculation

In this study, we used several gene signatures collected from public resources, including the Beltran NEPC Up gene signature, 76-gene AR-repressed signature, and Chen et al.’s RB1-loss signature [45,46]. The signature genes are listed in Appendix A. Summed z-score or ssGSEA are applicated to calculate signature activity. FPKM gene expression values were used as input of gsva function implemented in GSVA R package, using ‘ssGSEA’ method to calculate each sample’s signature score.

### 4.9. Clinical Data Analysis

The Abida and Alumkal cohort clinical data were utilized for analysis of prognostic features. Kaplan–Meier curves were estimated using the survfit function implemented in survival R package and plotted using survminer R package. Tumors were stratified in two ways. The first was into two groups of the top decile of ID2 Up signature scores versus the remainder, and the second was into median of ID2 signature score. The log-rank test was used to test for differences between survival curves.

### 4.10. qRT-PCR

Total RNA was extracted from cells using Tissue RNA Purification Kit (EZBioscience, Roseville, MN, USA), and 1 μg was reverse-transcribed using PrimeScript™ RT Master Mix (TaKaRa, Tokyo, Japan). Real-time PCR was performed using ChamQ Universal SYBR qPCR Master Mix (Vazyme, Nanjing, China). Target gene expression was normalized to GAPDH levels in three experimental replicates per sample. For primer sequences, please see Appendix A.

### 4.11. Overexpression

The CDS sequence of ID2 was obtained through NCBI, followed by amplification using the Hieff CloneTM One Step Pcr Cloning Kit (YEASEN cat# 10911ES20). The target sequence was recombined with the double-digested vector. The recombinant plasmid was transferred into transiently transfected cells using lipo3000, and the medium was changed after 6 h. The transfection efficiency was detected after 48 or 72 h using WB or qPCR. For stable overexpression, lentivirus was packaged using ID2 overexpression vector, psPAX2, and pMD2G vector.

### 4.12. shRNA Knockdown

ID2-targeting short hairpin RNAs (ID2-shRNA) and a nonspecific control (scramble) were constructed using a PGMLV-SB3 RNAi lentiviral vector. Sequences of ID2-shRNAs are listed in Appendix A. Lentiviral vectors were co-transfected with psPAX2 and pMD2G vectors into HEK293T cells. Supernatants were collected at 24 and 48 h after transfection. For infection, 5 × 104 cells were seeded in six-well plate and infected with lentiviral supernatant on the following day. Lentivirus, plasmid, and siRNAs were purchased from Genomeditech, Shanghai, China.

## 5. Conclusions

In this study, we first acquired multiple public datasets, screened for potential AR-negative drivers, and validated ID2 at the cellular and tissue levels. Cellular functions revealed the potential of ID2 to promote AR-negative prostate carcinogenesis. Mechanistically, transcriptome sequencing analysis suggested that ID2 promotes the acquisition of lineage plasticity through the JAK/STAT pathway and FGFR pathway. By combining the transcriptomic data in this study and the public prostate cancer dataset, we generated an ID2 signature gene set to help determine the likelihood of progression to AR-negative prostate cancer and its prognosis in prostate cancer patients treated with ARPIs.

## Figures and Tables

**Figure 1 cancers-16-00392-f001:**
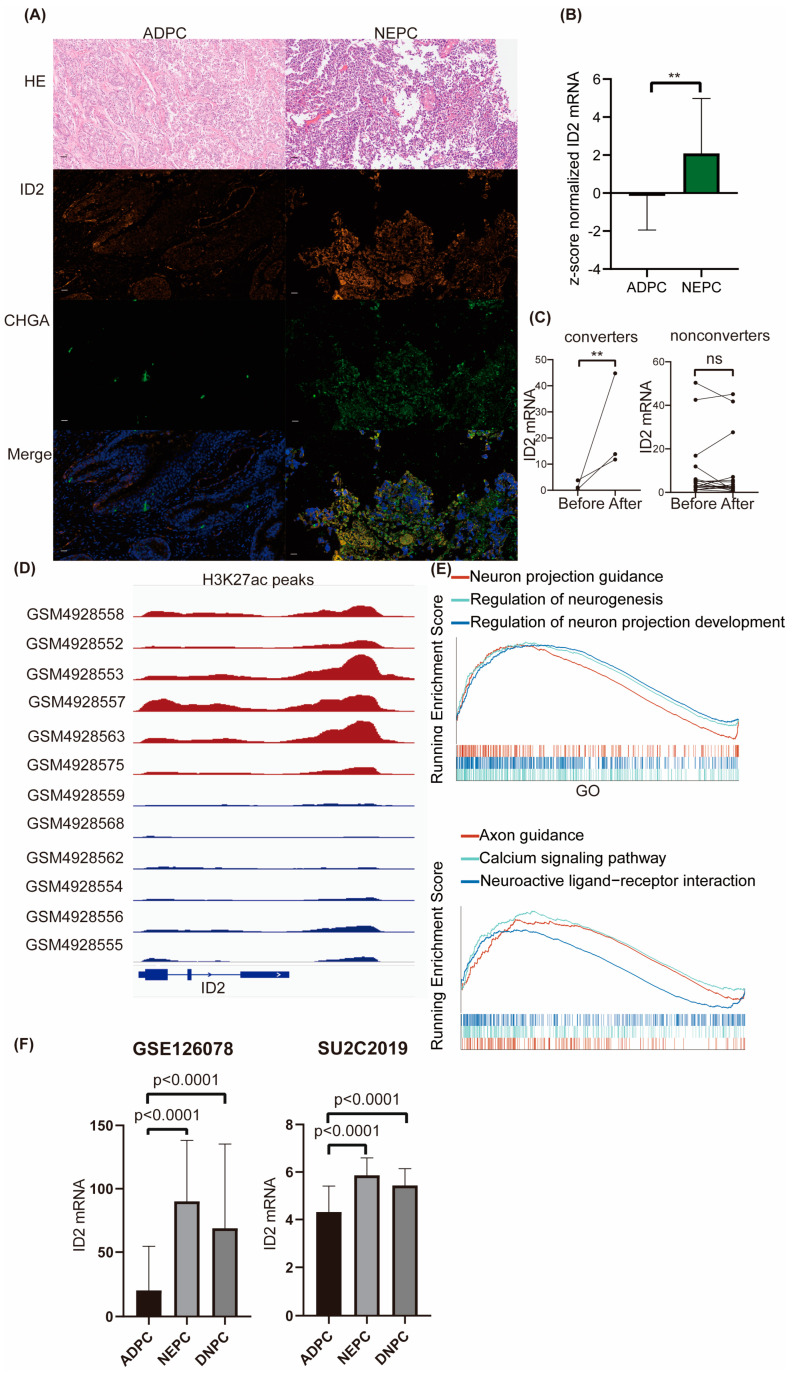
ID2 expression is upregulated in NEPC and DNPC. (**A**) Dual-tissue immunofluorescence analysis of ID2 and CHGA. HE represents hematoxylin and eosin staining. ADPC represents prostate adenocarcinoma, and NEPC represents neuroendocrine prostate cancer. Scale bar represents 20μm. (**B**) Z-score normalized mRNA expression of ID2 in Beltran’s dataset. Wilcoxon rank-sum test was used as statistical method; **: *p* < 0.01. (**C**) Relative mRNA expression of ID2 in Westbrook dataset; two-sample *t*-test was applied as statistical method; **: *p* < 0.01 ns: no significance. Converter represents samples that underwent NETD after enzalutamide, while nonconverters did not. (**D**) H3K27ac peak signals are around ID2 genomic loci in a series of LuCaP PDX models. Red represents NEPC, and blue represents ADPC. (**E**) GSEA was performed in ID2-high-expressing samples versus low-expressing samples from TCGA. Grouping method was the median expression of ID2 across all samples. Gene sets used in GSEA were imported from clusterprofiler R package (version 4.6.2). (**F**) Expression of ID2 in GSE126078 and SU2C2019 data. Mann-Whitney test was conducted to determine whether differences between groups are significant or not.

**Figure 2 cancers-16-00392-f002:**
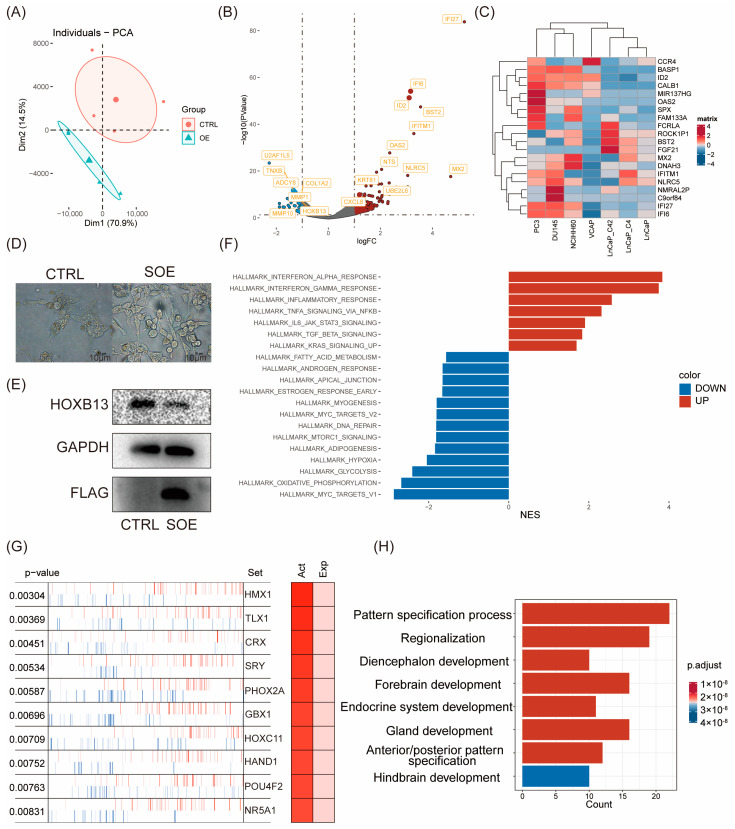
Transcriptomic reprogramming and pathway analysis upon ID2 overexpression. (**A**) Principal component analysis was performed comparing transient ID2-overexpressing PC3 cells and empty vector as control. (**B**) Volcano plot of differentially expressed genes (DEGs). |logFC| > 1 and *p* < 0.05 were defined as DEGs. (**C**) Top 20 DEGs across a panel of prostate cancer cell lines. (**D**) Morphological changes after a month of stable overexpression of ID2 in LNCaP cells. SOE represents stable overexpression. (**E**) Immunoblot of HOXB13 for the comparisons of ID2 ectopic expression versus control LNCaP cells. Three biological replications of the experiment were performed. *p* < 0.05; *t*-test was used to determine whether the differences were significant or not. (**F**). GSEA was performed to identify activated or deactivated pathways after ectopic expression of ID2 in PC3. Hallmark pathway gene sets from MSigDB were used as inputs. (**G**) Master regulator analysis of ID2-overexpressing PC3 cells. Master regulator analysis identifies top activated cells between ID2-overexpressing and control PC3 cells. Activity scores (**right**) and *p*-values (**left**, calculated using a gene shuffling test of the enrichment scores) were generated in the VIPER R package. (**H**) Functional enrichment analysis of activated master regulator after ectopic expression of ID2. The original western blot of Figure 2E is in Appendix A.

**Figure 3 cancers-16-00392-f003:**
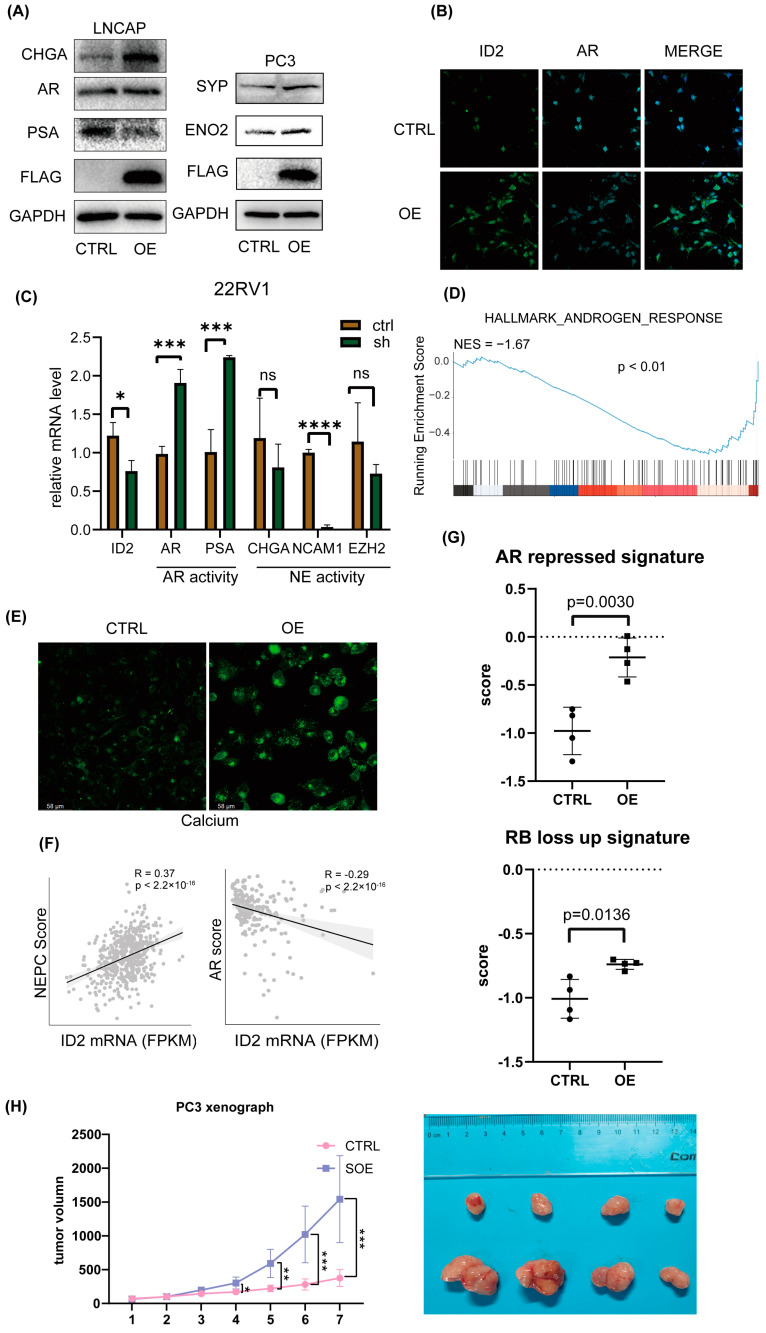
ID2 attenuates AR signaling and promotes NETD. (**A**) Immunoblot of indicated protein in transient ID2-overexpressing LNCaP or PC3 cells. (**B**) Immunofluorescence of ID2 and AR in ID2-overexpressing C42B cells. (**C**) Transcript levels of ID2 mRNAs were measured using qRT-PCR in ID2 knockdown and control 22RV1 cells. Significance was determined using Student’s *t*-test, and data are presented as mean ± SEM (*n* = 3 replicates per group). ***: *p* < 0.001. (**D**) GSEA of hallmark androgen response in ID2-overexpressing PC3 cells. (**E**) Calcium ion fluorescent probe imaging in ID2-overexpressing C42B cells. Scale bars: 58 μm. (**F**) Pearson correlation analyses of AR score or NEPC score and ID2 mRNA level. (**G**) Activity of AR-repressed signature or RB-loss up signature in ID2-overexpressing or control PC3 cells. (**H**) Tumor volume growth curve of ID2-overexpressing PC3-cell xenograft in nude mice. * *p* < 0.05, ** *p* < 0.01, *** *p* < 0.001, **** *p* < 0.0001, ns: no significance. The original western blot of Figure 3A is in Appendix A.

**Figure 4 cancers-16-00392-f004:**
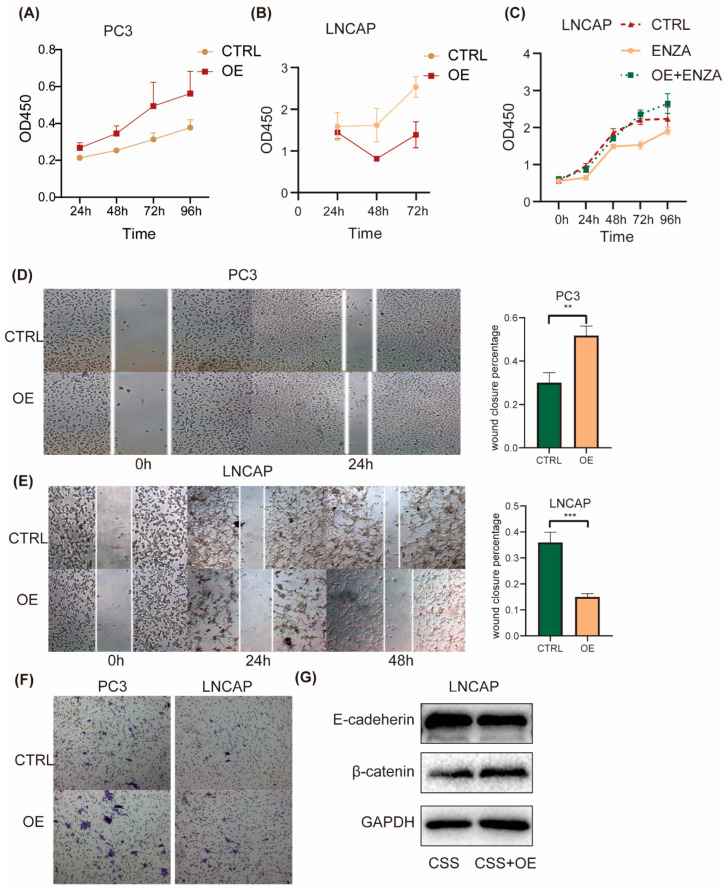
ID2 enhances PCa cell aggressiveness. (**A**–**C**) Growth of indicated cells with the overexpression of ID2 or with the empty vector as a control (CTRL) was analyzed using a CCK8 kit at different time points. (**D**,**E**) Wound-healing assays were performed with ID2-overexpressing cells or empty vector as a control at the indicated times; representative wound-healing images are shown in left panel, and percentage of wound closure is calculated in right panel. *p* < 0.01 when compared with the control group. Scale bars: 50 μm. (**F**) Transwell invasion assays were performed with PC3 cells overexpressing ID2 or control. (**G**) Immunoblot of E-cadherin and β-catenin in LNCaP cells with stable ID2 overexpression for a month or a control virus. ** *p*< 0.01, *** *p* < 0.001. The original western blot of Figure 4G is in Appendix A.

**Figure 5 cancers-16-00392-f005:**
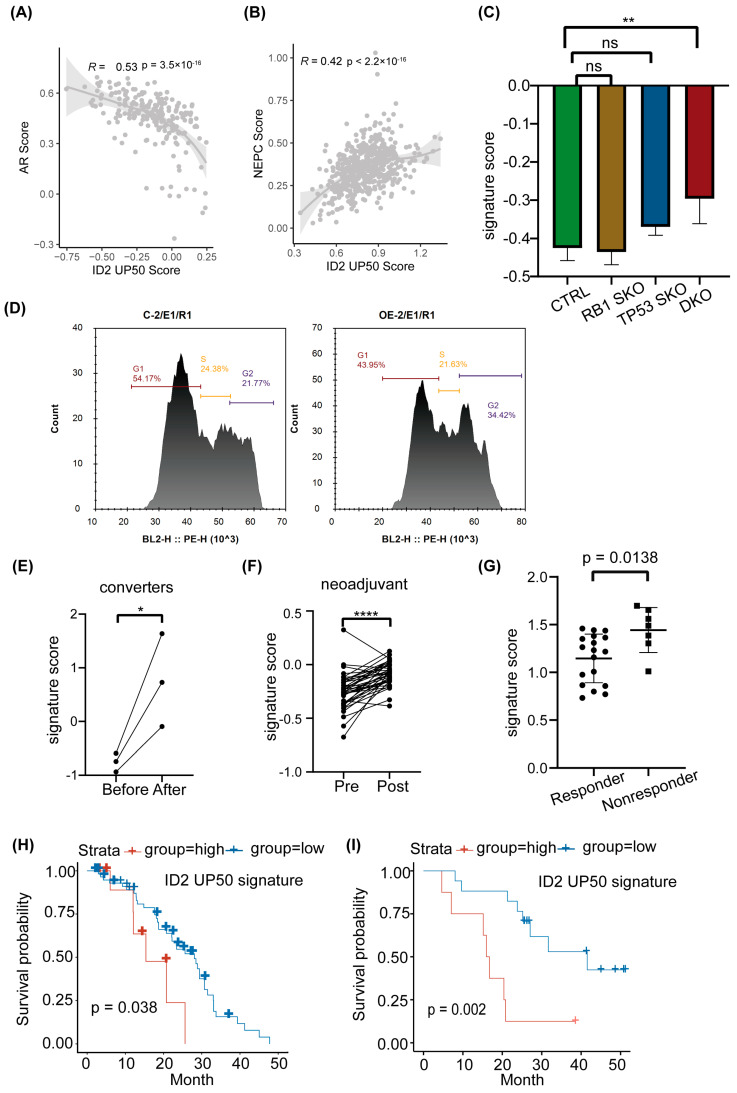
ID2 UP50 signature generation and validation of its effectiveness. (**A**,**B**) Pearson correlation analyses of AR score or NEPC score and ID2 UP50 signature score. Signature score was calculated from single-sample enrichment analyses, implemented by gsva in the GSVA R package. (**C**) ID2 UP50 signature score in genetically modified LNCaP cells. SKO represents single knockout. DKO represents TP53 and RB double knockout. ** *p* < 0.01. ns: no significance. (**D**) Cell cycle analysis using flow cytometry for the comparison of ID2-overexpressing and control PC3 cells. (**E**–**G**) Activity analysis of ID2 UP50 signature in different clinical samples. (**E**) Paired clinical samples that underwent NETD before and after enzalutamide treatment. (**F**) Paired clinical samples that underwent neoadjuvant enzalutamide therapy before surgery. (**G**) Alumkal’s clinical samples grouped by whether they responded to enzalutamide therapy. A decrease in PSA levels of less than 50 percent was defined as a nonresponder, otherwise a responder. Significance was determined using Student’s *t*-test. *: *p* < 0.05; ****: *p* < 0.0001. (**H**,**I**) Kaplan–Meier curves stratified using ID2 UP50 signature score. Tick marks indicate censoring events. *p*-values were determined using the log-rank test to compare outcome measures between nonresponders and responders.

**Figure 6 cancers-16-00392-f006:**
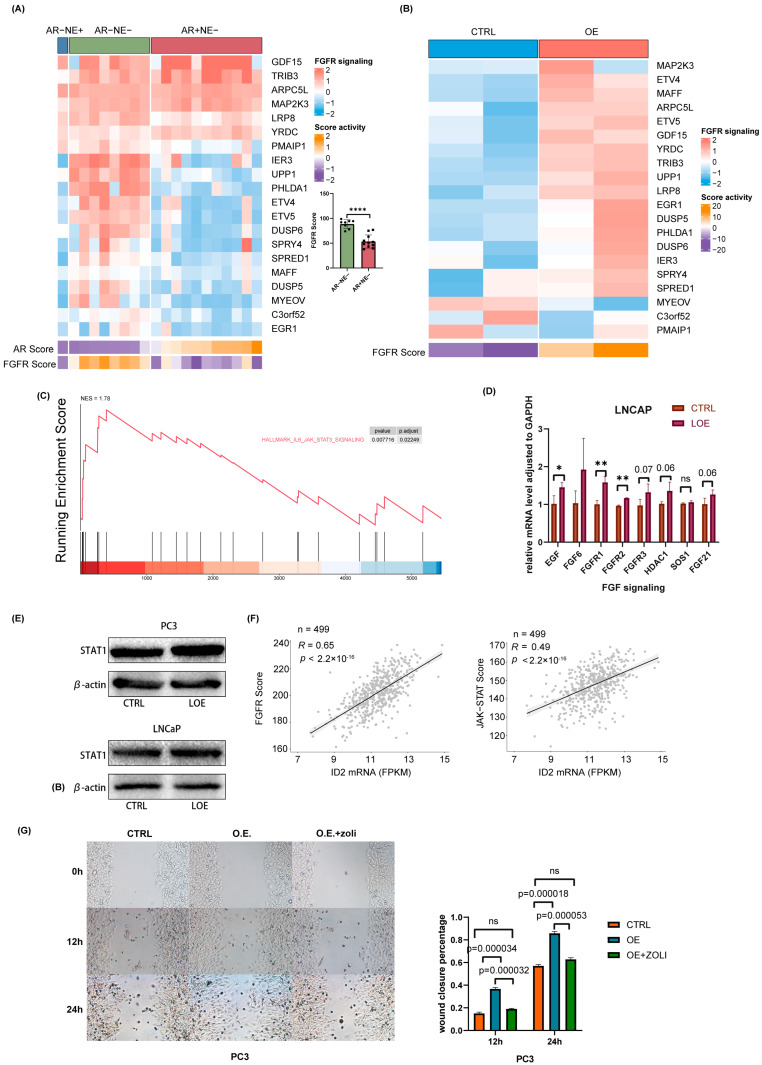
ID2 activates FGFR signaling and JAK-STAT signaling to promote lineage transition. (**A**) RNA-sequencing-based measurements of transcripts comprising FGFR-signaling-associated genes in a panel of 20 PCa cells. Cells were clustered into three groups based on their AR activity scores and NEPC scores. Each dot represents a prostate cancer cell line. (**B**) Measurements of the effect of ID2 on FGFR-signaling-associated genes in PC3 cells. (**C**) GSEA of IL3-JAK-STAT3 was performed after ID2 overexpression in PC3 cells. (**D**) qPCR of FGFR-signaling-associated genes in ID2-overexpressing LNCaP cells. (**E**) STAT1 immunoblots of cell lysates from ID2-overexpressing LNCaP and PC3 cells. Three biological replications of the experiment were performed. *p* < 0.05; *t*-test was used to determine whether the differences were significant or not. (**F**) Correlation of ID2 transcript levels and FGFR pathway activity and JAK-STAT activity scores assessed in 499 primary PCa using RNA-seq. Pearson’s correlation coefficient and *p*-value are indicated on each plot. (**G**) On the left panel is a representative image of a PC3-cell scratch assay. O.E. indicates ID2 overexpression, and zoli indicates zoligratinib, an inhibitor of the FGFR pathway. Cells were photographed at 0 h, 12 h, and 24 h. Scale bar: 100 um. On the right is the wound closure rate at different time points after scratching. The wound closure area was obtained using Image J (version: Image J 1.53e) analysis, and the experiment was performed with 3 biological replications. *t*-test was used to determine whether the difference between groups was significant or not. * *p* < 0.05, ** *p* < 0.01, **** *p* < 0.0001, ns: no significance. The original western blot of Figure 6E is in Appendix A.

## Data Availability

All the data generated in this study can be accessed upon request.

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
