# Peer review of "ID2 Promotes Lineage Transition of Prostate Cancer through FGFR and JAK-STAT Signaling"

_cancers, 2024, doi:10.3390/cancers16020392_

Round 1
Reviewer 1 Report
Comments and Suggestions for Authors
Summary:
In this study, the authors investigated the role of ID2 in prostate cancer (PCa) lineage transition, tumorigenesis, and resistance to enzalutamide. They found that the RNA and protein expression of ID2 was increased in AR-null PCa and that the pathways activated by ID2 were associated with neurodevelopment. ID2 had a higher signature score in neuroendocrine prostate cancer (NEPC) samples than adenocarcinoma of the prostate (ADPC), which correlated with an unfavorable clinical outcome for PCa patients. Moreover, the authors showed that ID2 promoted PCa tumorigenesis in vitro and in vivo. ID2 overexpression increased the activation of JAK-STAT and FGFR signaling where the latter was found to be significantly activated in AR-null PCa cells including double-negative prostate cancer (DNPC) and NEPC.
Comments to authors:
Although this paper is interesting, especially in terms of PCa linage transition, the authors should address the following:
- The title can be improved as it is not clear what ID2 is
- Abbreviations used in the article must be revised as some abbreviations were put without introducing them, especially in the abstract
- The last sentence in the abstract “Our study indicates that ID2 promotes the acquisition of a stem-like phenotype in adenocarcinoma cells, leading to resistance to ADT and next-generation ARPIs in prostate cancer” is an overstatement as this was not adequately or extensively proven in the article
- It is recommended to re-write the introduction as it is not clear and does not serve the purpose of this study. ID2 which is the focus of this research was introduced at the end and still, it was not clear what it is. A set of markers was introduced without clarification of their roles or their full names. The aim of the study at the end of the introduction needs re-writing.
- Regarding 22RV1, is it a hormone-resistant cell line?
- Regarding dual immunofluorescence analysis of ID2 and CHGA, stating that “with the former being positive for CHGA and strongly positive for ID2, whereas in the latter, it was negative for CHGA and weakly positive for ID2 should be backed up by quantification of the signal intensities
- For figure S1b, the key or legend to the figure is not put
- RNA sequencing was mined from 21 paired samples after enzalutamide treatment, of which 3 samples had histologically confirmed trans-differentiation to neuroendocrine prostate cancer (NEPC), while the remaining 18 did not develop NETD. Data on ID2 expression was built on 3 samples out of 21 while the remaining 18 had fluctuating results. Could having a larger sample size with NETD also give conflicting results with different trends?
- For Figure 2a, in the results section “samples with ID2 overexpression” showed differences from the control group in terms of principal components. Looking at the figure legend, only PC3 cells overexpressing ID2 were used. This should be clarified in the results section
- In the “Transcriptomic reprograming and pathway analysis upon ID2 overexpression” section, line 9, the authors used the term “moderately higher expressed”. Is it statistically significant?
- For western blot results, densitometry quantification should be implemented. Also, it is not mentioned how many experiments or replicates these results were based on
- Choosing western blot assay for some cell lines versus immunofluorescence assay for another is not clear (see Figure 3A and 3B). This point is also applicable to other assays/analyses. Why were these cell lines chosen specifically, in particular PC3, and switching between the cell lines and using them on different assays is not clear
- Again, for Figure 4C, is the difference between LNCAP cells and LNCAP overexpressing ID2 treated with enzalutamide significant?
- It is recommended to rephrase this phrase “However, the results of invasion assay showed consistency: Although ID2 inhibited the proliferation and migration of LNCaP cells, it enhanced its invasion ability (Figure 4e), which was consistent with the results of invasion assay of PC3”
- Although the discussion section clarifies the whole obtained results of this research, it is only a summary of the results. It is recommended to re-consider it while emphasizing the significance of the results in terms of current research
- In the discussion, it was mentioned that “Using immunoprecipitation, we confirmed that ID2 can interact with RB1, which may lead to its inactivation”. Yet, immunoprecipitation assay was not mentioned in any of the results or the methods sections
- In the methods, “cell culture” section, it is not mentioned how 22RV1, C42B, and PC3 cells were maintained
- It is recommended to add a section about the statistical analysis in the methods section
- The English writing throughout the article is recommended to be revised
Comments on the Quality of English LanguageThe English writing throughout the article is recommended to be revised
Author Response
Thank you for the comments. All the points you listed is now replied one by one in attached cover letter and revisions have been highlighted in red in the revised manuscript.
Please see the attachment.

Reviewer 2 Report
Comments and Suggestions for Authors
In this original research article by Zhang et al, the role of ID2 in prostate cancer is investigated. The primary conclusions are that ID2 expression and activity induces transcriptional and functional changes in PCa cell lines to promote cellular plasticity, AR-indifference and neuroendocrine differentiation through FGFR and JAK/STAT signaling. To support these claims, a variety of molecular and bioinformatic approaches were employed and potentially interesting outcomes were identified. However, the manuscript lacks several controls and orthogonal assays to validate claims and significant work will be required accurately assess author claims and interpretations regarding ID2 activity in prostate cancer. Below are comments and suggestions:
Major:
-The authors sate that ID2 is upregulated in both NEPC and DNPC (Abstract, Results section heading (Page 2) and Figure 1 legend) but it is not clear what data represents DNPC in this manuscript. PC3 cells are stated in the manuscript as “NE like” (Page 4, paragraph 2) and are shown to express SYP and other NE markers in subsequent figures. Further, analysis of "DNPC" cell lines in Figure 6 identifies FGFR signals but does not directly correlate these cell lines to ID2 expression or pathway activity. Comparisons between ID2 expression in NEPC and adenocarcinoma patient datasets (Beltran) are conducted, but similar analyses including DNPC patients are missing. Please also conduct analysis in DNPC patient samples with publicly available RNAseq profiles to support this claim (GSE126078 or SU2C data already reported in this manuscript).
-The majority of the bioinformatic and functional analyses in model systems are conducted in ID2-overexpressing PC3 and LNCaP cells. However, the authors have shown high expression of ID2 transcript levels in PC3 cells compared to other prostate cancer cell lines at baseline (Supplementary Figure 1A, Figure 2C) and do not quantify ID2 protein in PC3 and LNCaP cells either at baseline or after ID2 ectopic expression. FLAG protein expression is used as a surrogate for ID2 protein overexpression on Western Blot (Figures 2 and 3), but published literature shows that ID2 primary antibodies for WB are readily available (example - PMID: 35422476). Further, they show ID2 protein detection by immunofluorescence in C42B cells (Fig 3B), but do not use image quantification tools or recapitulate IF data in other manipulated cell lines to determine if ID2 protein is appreciably expressed ectopically or knocked-down compared to control cells.
-The authors make several claims regarding ID2 activity on AR transcriptional programs throughout the manuscript, but do not sufficiently characterize functional consequences of ID2 manipulation on AR activity in any single AR-expressing cell line. For LNCaP ID2-OE cells, AR protein expression is reported to be decreased but does not appear to be altered (Figure 3A) and orthogonal assays are not used to support the claims of broad decreases in AR-activity. Does qPCR confirm that AR-regulated genes are repressed in LNCaP ID2-OE cells relative to control LNCaP cells? Does RNAseq and GSEA of LNCaP ID2-OE cells show alterations in AR regulated pathways? Many assumptions about the ID2-AR relationship in LNCaP cells are inferred from patient datasets or data derived from other AR+ cell lines. These assumptions must then be carried forward to explain results of the proliferation and wound healing assays for LNCaP ID2-OE cells in Figure 4 .
-The conclusion that ID2 overexpression increases invasion and EMT programs in PC3 and LNCaP cells is derived from anecdotal data (Figure 4F and G). The number of invading cells from transwell assays are not quantified and statistically compared (Figure 4F). Western blots comparing beta-catenin and e-cadherin between ID2-OE and control cells are not quantitative and do not appear to be significantly altered (Figure 4G). Were these assays conducted in triplicate? Does the expression of vimentin or MMPs change with ID2 expression? Are EMT/Invasion programs altered in the RNAseq of PC3 ID2-OE cells? Parallel qPCR or transcriptome analysis of EMT/Invasion/metastasis genes in LNCaP ID2-OE cells is required to validate involvement of these programs.
-FGF and JAK/STAT signaling are identified from transcriptional data and the authors claim that these pathways drive ID2-induced lineage plasticity in PCa. However, western blots from ID2 overexpressed lines do not show obvious changes in STAT expression (Figure 6E). Additionally, FGF and JAK/STAT pathways were not experimentally validated in their ID2-OE models to show that perturbations (i.e. pharmacological inhibitors, siRNAs etc.) mitigate the functional or transcriptional consequences of ID2 overexpression.
Other:
-Figure 1A, using a single patient specimen for each subtype is not statistically relevant and authors are making broad claims that ID2 and neuroendocrine markers are co-expressed. The authors do not discuss if there is the potential for tumor heterogeneity or if some NEPC tumors may also have little or no ID2 expression.
-Please represent Figure 1G, Figure 3G and Figure 5G as a scattered boxplot with each tumor as a point on the graph to show the distribution of patient data more transparently.
-Patient specimens (Figure 1A) - no methodology on sample acquisition or how the samples were stained and visualized are stated. Is the NEPC biopsy a de novo primary specimen or t-NEPC/metastases? Are the specimens treatment naive? How were the tissues prepared for immunofluorescence and what antibodies were used?
-Figure 1D - the LuCaP PDX models are well characterized and well known prostate cancer preclinical models. Please label the H3K27ac ChIPseq tracks to the corresponding LuCap models. RNAseq and Microarray data for many of the LuCaP lines are publicly available (eg. GSE126078). Does enriched H3K27ac ChIPseq in PDX models correlate with ID2 expression? Do NEPC Models have higher expression of ID2 compared to adenocarcinoma LuCaP models?
-Please clearly explain how Figure 1E supports your hypothesis. The authors suggest that ID2 expression is upregulated in NE tumors, however, leading edge genes and enrichments scores for neuroendocrine gene sets are highest in the ID2_low tumors.
-ID2 is reported to promote neuroendocrine transcriptional programs in ID2-OE cells. However, alternative approaches to validate the observations are not well described. Given that parental PC3 cells express ID2 mRNA, does ID2 knockdown in PC3 parental cells decrease baseline NE gene expression? Alternatively, does pharmacological inhibition of ID2 or ID2 pathway (homoharringtonine, helichrysetin, etc.) phenocopy ID2 knockdown in PC3, NCIH660 or 22rv1 cells?
-Figure 2F – It is not clear from the manuscript or figure legend if this data is derived from PC3 cells with ID2-OE or from another source.
-Figure 3G is not clear, the legend states that this is data from AR-negative PC3 cells with ID2 OE and the manuscript states it is patient data.
-Figures 4D, E and F are not clearly labelled or explained in the figure legend. The reader must assume which images represent control and experimental conditions. Timepoints for images taken “...at the indicated times” for 4D and 4E are not specifically described.
Comments on the Quality of English Language
Acceptable. No major editing needed.
Author Response
Thank you for your careful and conscientious review of the manuscript. Based on the few points you raised, I have revised the Manuscript and highlighted it in red. In addition, I wrote a cover letter, performed additional bioinformatics analysis in order to answer the major points in a targeted manner.
Please see the attachment

Reviewer 3 Report
Comments and Suggestions for Authors
This is an well conducted, comprehensive study regarding the role of ID2 promotes lineage transition of prostate cancer, especially in NEPC.
some minor comments:
1: a list of abbreviation is needed , such as ID2, ARPIs, DNPC, CCLE, H3K27ac, GSEA, etc, this will made the readers easily to follow
2. for resistance to enzalutamide, there are several mechanisms, NEPC is one of them, how about the proportion? 20%, 30% ? prostate cancer become resistance after enzalutamide is due to NEPC?
3: reviewer are curious to known: the phenomenon of ID2 promote lineage transition of prostate cancer into NEPC through FGFR and JAK-STAT signaling. is a common mechanism, or only for a small set of NEPC in human prostate cancer?
4. though these results of current study: ID2 promotes the acquisition of a stem-like phenotype in adenocarcinoma cells, leading to resistance to ADT and next-generation ARPIs in prostate cancer.
Question : how to prove the stem-like phenotype cell is NEPC or DNPC? the author did not answer the question
Comments on the Quality of English Language
The quality of English is well in this manuscript
Author Response
Thank you for the comments. All the points you listed is now replied one by one and revisions have been highlighted in red in the revised manuscript.
Please see the attachment

Round 2
Reviewer 1 Report
Comments and Suggestions for Authors
Please see my previous comments.
Comments on the Quality of English LanguageNeeds extensive editing